# The Safety and Impact of Raising the Urine Culture Reporting Threshold in Hospitalized Patients

**DOI:** 10.3390/jcm11237014

**Published:** 2022-11-27

**Authors:** Ohad Gabay, Tal Cherki, Gal Tsaban, Yoav Bichovsky, Lior Nesher

**Affiliations:** Infectious Disease Institute, Internal Medicine Division, Soroka Medical Center, Faculty of Health Sciences, Ben-Gurion University of the Negev, Beer Sheba 84101, Israel

**Keywords:** asymptomatic bacteruria, antimicrobial stewardship, urinary tract infection, diagnosis

## Abstract

**Objective**: To assess the impact of changing the reporting threshold policy of positive urine cultures in hospitalized non-pregnant adults from 10^4^ CFU/mL to 10^5^ CFU/mL on the unwarranted use of antibiotics and patient safety. **Setting**: A 1100-bed tertiary-care hospital in southern Israel. **Methods**: As an intervention, we changed urine culture reporting policy for patients admitted to general medical wards. If culture grew ≥10^5^ CFU/mL, it was reported with pathogen and antibiotic susceptibility data, if it grew ≤10^4^ CFU/mL, it was reported as “low growth". The withheld information was available upon request. We retrospectively collected data on all patients in a four-month period following the intervention and report using STROBE guidelines. **Results**: 7808 patients were admitted, in whom 3523 urine cultures were obtained. A total of 496 grew a pathogen, 51 were excluded (*candida* spp. positive, history of urinary surgery, obtained from catheter). A total of 300 were reported as positive and 145 were reported as low-growth. A higher rate of patients in the low-growth group were not treated with antibiotics 45/145(31%) vs. 56/300(18.7%) in the positive group *p* = 0.015 and the antibiotic duration of treatment was shorter by day 5 (IQR 0.9) vs. 6 (IQR 0.9) *p* = 0.015. No between-group difference was observed in recurrent admission rates, pyelonephritis within 30 days, bacteremia or all-cause mortality. **Conclusions**: Changing the reporting threshold of positive urine culture results from 10^4^ CFU/mL to 10^5^ CFU/mL in hospitalized patients reduced the number of patients who were unnecessarily treated for asymptomatic bacteriuria without negatively impacting patient safety. We urge microbiological laboratories to consider this change in threshold as part of an antimicrobial stewardship program.

## 1. Introduction

### 1.1. Background Rationale

Urinary tract infection (UTI) is the most common and over-diagnosed bacterial infection in adults [1]. Asymptomatic bacteriuria (ASBU) is frequent in older adults, both in the community and in health care institutions. The prevalence of ASBU ranges from 2% to 10% in the community and can be as high as 40–50% in long-term care facilities [1]. The presence of a positive urine culture even when a different diagnosis is considered (infectious or non-infectious) creates a challenging diagnostic dilemma for clinicians and frequently leads to an increased prescription of antimicrobials, particularly among patients who have a fever from a different cause [2].

The identification of a pathogen causing a urinary tract infection is important, as it can assist in guiding and narrowing antibiotic therapy. Culturing a suprapubic aspiration is considered the gold-standard for diagnosing UTI, where any number bacteria-forming units is considered pathogenic; however, it is rarely performed in clinical practice, especially in the adult population [3]. Obtaining a clean catch of midstream urine after cleansing of the first urine of the day is considered an appropriate alternative and is the most common method of acquiring urine culture in adult population [4]. In reality, most urine cultures are performed whenever patients approach medical services and many are performed without cleansing or proper technique, leading to a certain degree of artifactual contamination [5,6].

The standard threshold for bacterial growth on a properly collected urine sample that is considered to represent true bladder bacteriuria is ≥10^5^ colony-forming units (CFU)/mL. (17) This threshold has been challenged and it has been recognized that, in certain conditions, a lower threshold is appropriate, especially if patients were pretreated with antibiotics [2,7,8,9,10]. This led the value of cultures that are reported as positive being lowered to 10^4^ CFU/mL, which, in turn, led to an increase in the number of positive urine cultures, but may lead to overdiagnosis of ASBU [2,10,11]. In recent years, due to a growing concern regarding teh emergence of antimicrobial resistance secondary to increased antibiotic stress and unnecessary treatment of ASBU, it has been suggested to increase the threshold back to 10^5^ CFU/mL [2,11,12].

### 1.2. Objectives

We hypothesized that changing the reporting threshold policy of positive urine cultures in hospitalized non-pregnant adults from 10^4^ CFU/mL to 10^5^ CFU/mL will result in a reduction in unwarranted antibiotics’ usage, especially in those with ASBU, without compromising patient safety.

## 2. Methods

### 2.1. Design

This is a retrospective study collecting data on all patients admitted to the general medicine wards of Soroka University Medical Center (SUMC) between August and November 2017. We compared the patients in whom the urine culture grew 10^4^ CFU/mL and were reported as “low growth” without a specific pathogen or antibiotic sensitivity pattern to urine cultures that grew 10^5^ CFU/mL and were reported as positive with pathogen identification and antibiotic sensitivity pattern. Approval and waiver of informed consent for this study was obtained from the institutional ethics committee and this report utilizes the standards for transparency reporting set by the STROBE guidelines [13].

### 2.2. Setting

SUMC is a 1100-bed tertiary hospital in southern Israel serving approximately one million people; it is the only hospital in a 100 km radius. Standard of care at SUMC for patients admitted with fever and those that are suspected to have an infectious process includes performing a urine culture. Urine cultures are performed in the local microbiological laboratory 5 days a week, positive cultures are identified and antimicrobial susceptibilities determined using VITEK 2 bioMérieux Clinical Diagnostics, all according to CLSI standards [14]. Final results are electronically reported through the patients electronic medical record.

### 2.3. Participants

Our protocol included all adults (≥18 years old) patients who were admitted to SUMC general medicine wards between August and November 2017 who grew a positive urine culture for a single bacterial pathogen. We excluded all patients in whom the urine was obtained from a urine catheter, nephrostomy tubes, polymicrobial culture, as well as pregnant patients and those with a history of urinary tract surgery; in these patients, the laboratory reported all pathogens and sensitivity data regardless of CFU/mL count.

### 2.4. Intervention

Starting July 2017, all positive urine culture results of patients admitted to general medical wards were reported in one of two ways: cultures that grew at least 10^5^ CFU/mL were reported in the regular manner including the pathogen name and antimicrobial susceptibility pattern; cultures that grew 10^4^ CFU/mL or less were reported as “positive low growth”, without the pathogens name or susceptibility pattern. The withheld information was available to the medical staff upon request.

### 2.5. Data

We collected data on all patients who had a positive urine culture for a four-month period following initiation of the intervention. Data were collected by manually reviewing the patient’s electronic medical records and recording all data on the hospitalization in which the positive culture grew, as well as a 90-day follow-up period.

### 2.6. Outcomes

Definitions of outcomes can be found in Table 1; the outcomes included both microbiological failures determined as repeat urine culture or bacteremia with the same pathogen within 30 days. Other outcomes included all-cause mortality, readmission within 30 days, and treatment with antibiotics and *C. difficle.*

### 2.7. Statistical Methods

Our primary aim in this study was to compare antibiotic usage and duration in adult, non-pregnant, hospitalized population in SUMC before and after the implementation of laboratory threshold changes to reporting a positive urine culture. Sample size calculations for this study are provided in the supplementary section. Continuous variables are presented as means ± standard deviations or as medians (interquartile ranges) as appropriate. Categorical variables are presented as percentages. Comparisons between groups were performed using independent *t*-test or Mann–Whitney test for continuous variables, as appropriate, and Chi-square test for categorical variables. A *p*-value ≤ 0.05 was considered statistically significant. All analyses were performed using SPSS version 21 (IBM, Chicago, IL, USA).

## 3. Results

Between August and November 2017, 7808 patients were admitted to the general medical wards; in those patients, 3523 (45.1%) urine cultures were collected, of which 496 (14.1%) were positive, 51 were excluded as they were samples from urine catheters, patients with a history of urinary surgery or grew candida. Of the 445 remaining cultures, 300 were reported as a positive urine culture (RP) with the identification of pathogen and antibiotic sensitivities, and 145 were reported only as low growth (LG); pathogen identification and susceptibly patterns withheld and reported upon request. (Figure 1) Baseline patient characteristics are shown in Table 2; no statistically significant differences were observed between the groups. A high rate of diabetes (~30%) was observed in both cohorts as well as high rates of chronic renal failure (17%). About half of the patients presented with fever and leukocytosis. In the RP group, 17 patients (5.7%) presented with hypotension, of which 16 had bacteremia congruent with and similar to their reported bacteriuria. In the LG group, 11 (7.7%) presented with hypotension, 6 of which had bacteremia congruent with and similar to their reported bacteriuria. Microbiological data on the urine cultures are presented in Table 3. The majority of identified urinary pathogens were gram-negative enteric; higher rates of enterococcus were observed in the LG group as compared to the RP [19 (13%) vs. 12 (4%), *p* < 0.001, as well as pseudomonas, with 13 (9%) vs. 12 (4%) *p* = 0.028, respectively.

The empirical antibiotics treatment rates were high in both PG and LG groups; however, they were significantly higher in the PG group, 244/300(81.3%) and 100/145(69%) *p* = 0.015, respectively. Comparing the non-treated patients in both groups (56 vs. 45 pts) the recurrent hospitalization rates tended to be higher but not significant in the 10^5^ group 7 (12.5%) vs. 1 (2.2%) *p* = 0.057, and no difference was found in the recurrent bacteremia rates 6 (10.7%) vs. 4 (8.8%) *p* = 0.756 or in the mortality rates 5 (8.9%) vs. 3 (6.6%) *p* = 0.418 (Table 4).

The antibiotic duration of treatment was shorter by a day 5 (IQR 0.9) and 6 (IQR 0.9) in the LG and PG groups: respectively *p* = 0.015. No between-group difference was observed in recurrent admission rates, pyelonephritis within 30 days, bacteremia or all-cause mortality (Table 5). All deaths observed in these cohorts occurred in the sentinel hospitalization. In both groups, none of the patients who did not receive antibiotics died or developed complications such as pyelonephritis, bacteremia or recurrent admission. In the LG group, no patient suffered from pyelonephritis. One patient had bacteremia within 30 days, which was from a different pathogen determined to be not of urinary origin. Of the patients with a recurrent growth in urine culture 6 had the same pathogen and 4 presented with a different pathogen growth.

## 4. Discussion

Asymptomatic bacteriuria is well-known to be one of the drivers of unnecessary antibiotic therapy in patients. The threshold for reporting positive urine cultures was historically set at 10^5^ CFU/mL. Due to concerns regarding the underdiagnosis of UTI, especially in partially treated outpatient women, many laboratories lowered their reporting threshold to 10^4^ CFU/mL several years ago [4,15]. It was recently suggested that increasing the reporting threshold to 10^5^ will reduce unnecessary antibiotic treatment [11,12]. This intervention was set in place as part of a hospital-wide antibiotic stewardship program, which included other interventions, and we aimed to assess the impact and safety of this. In our cohort, only 50% of the patients had fever or objective signs of an infection; thus, in about 50% of the patients, a urine culture was performed due to either clinical suspicion of an infection or for unclear reasons. We found a high rate of antibiotic usage in both groups (81.3% and 69%). The physician’s decision to start antibiotics was primarily based on clinical presentation rather than culture results. Examining the RP group, 56 patients (18.7%) were not treated with antibiotics and the urine culture was deemed to be ASBU by the clinicians; this rate improved in the LG group to 45 patients (31%). We observe that changing the threshold for reporting helped clinicians to avoid unnecessary antibiotic therapy. A urine culture reported as positive, including the pathogen name and susceptibility pattern, is a strong influence on clinicians, forcing them to reconsider their diagnosis and determine that the patient has ASBU to withhold antibiotics. In the LG cohort, where the report was of a low growth without pathogen identification or susceptibility data, the percentage of patients determined to have ASBU increased; thus, the lack of information assisted the clinicians in determining that this was an ASBU without the need for antibiotic therapy.

Our cohort included only hospitalized patients, many of whom had significant comorbidities and were admitted to acute medical care wards. All patients suspected of having an active infection were started on antibiotics prior to culture results. The antibiotics were adjusted as required according to the culture results in both groups, as microbiological data was available upon demand to clinicians in the LG group. This fact helps to exclude patients with ASBU and increase the safety of the intervention. When analyzing patients who were not treated with antibiotics in both groups, which could either represent ASBU or an occult infection, none developed pyelonephritis, became bacteremic or died. These results are similar to previously published interventions of a pilot study performed by leis and colleagues [11]. When a clinician chooses to perform a urine culture without starting empiric antibiotics, his assessment of a pretest probability for infection is low, thus withholding empiric antibiotic therapy. A positive urine test result may alter this decision; however, we have shown that patients in the LG group were not adversely affected by the change in the reporting threshold. To strengthen this concept, a prospective randomized control trial, focused on the outcome of non-treated patients in both groups, is still needed.

The pathogens identified in both groups were mostly gram-negative enteric pathogens as is expected; however, we did see a larger percentage of *Enterococcus* spp. in the cultures that grew 10^4^ CFU/mL. This observation is similar to a previous report by Hooton and colleagues [4]. *Enterococcus* spp. Urine with low counts is considered to be of lower virulence and is usually not covered by empiric antibiotic therapy for urinary tract infection [16]. Our study demonstrated that, even though we observed a large percentage of untreated *Enterococcus* spp., we did not observe higher failure or recurrence rates in these patients.

There are several limitations to this study. Yhe first and foremost is its retrospective nature; this limits the available clinical information on these patients, and precludes understanding of the reasoning behind some of the urine cultures that were obtained. We do not have data regarding how often the physician in the LG group asked for full bacteriological results and antibiotic sensitivity. We have limited data on antibiotics taken prior to admission, and thus cannot discuss the impact of prior antibiotic therapy on culture results.

## 5. Conclusions

Changing the reporting threshold of positive urine culture results from 10^4^ CFU/ml to 10^5^ CFU/ml on hospitalized patients may reduce the number of unnecessarily treated patients for asymptomatic bacteriuria without negatively impacting patient safety. We urge microbiological laboratories to consider this change in threshold as part of an antimicrobial stewardship program. 

## Figures and Tables

**Figure 1 jcm-11-07014-f001:**
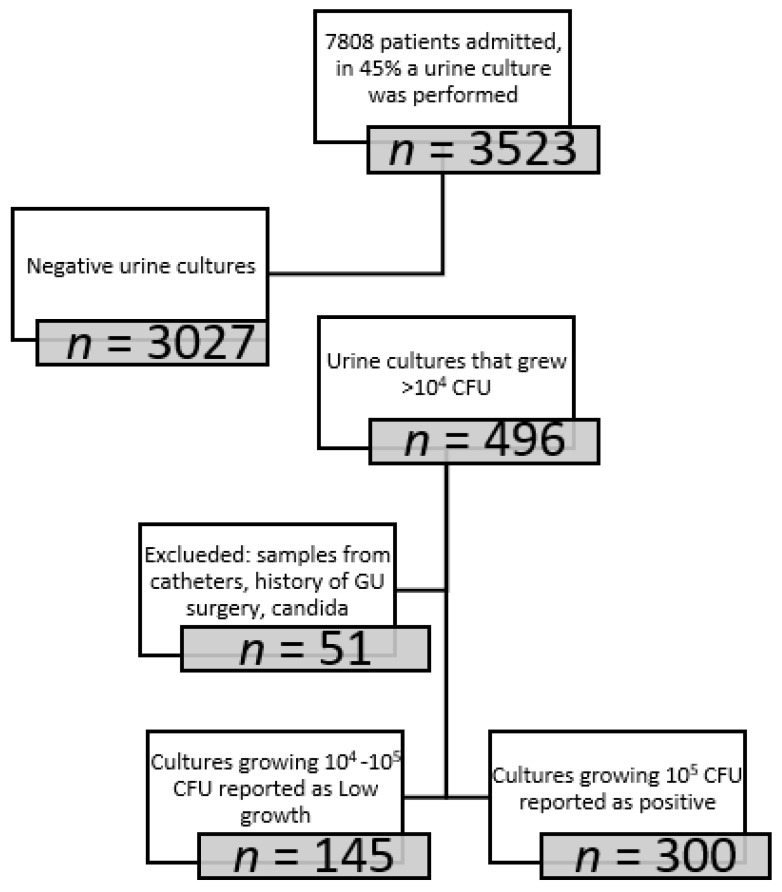
Flow chart of patients included in the study cohort who were admitted to the General Medical wards during a four-month period following an intervention to urine culture report policy.

**Table 1 jcm-11-07014-t001:** Definition of outcomes for the intervention.

Admission Duration	Length of Hospitalization in Which the Positive Urine Culture Was Identified
No antibiotics therapy	Patients in whom no antibiotic therapy was provided during the hospitalization or recommended on discharge
Antibiotic therapy duration	Length of antibiotic therapy during hospitalization
Recurrent admission 30 day	Patients who were readmitted for any cause within 30 days
positive urine culture 90-day same pathogen	A positive urine culture obtained within 90 days of the original culture and grew the same pathogen as the original culture
positive urine culture 90-day different pathogen	A positive urine culture obtained within 90 days of the original culture and grew a different from the original culture
Bacteremia within 30 day	Bacteremia within 30 days of the same pathogen that grew in the original urine culture
Mortality all cause 90 day	All-cause mortality within 90 days
*C. difficile*	Patients who developed *C. difficle* infection within 3 months of the positive culture

**Table 2 jcm-11-07014-t002:** Baseline patient characteristics comparing patients with urine culture that were reported as low growth to patients that were reported as positive.

	Reported Low Growth *n* = 145 *n* (%)	Reported Positive *n* = 300 *n* (%)	*p* Value
Gender female	97 (68.3)	217 (72.3)	0.384
Age (standard deviation)	58 ± 23	59 ± 24	0.578
Diabetes mellitus	45 (31.7)	91 (30.3)	0.773
Chronic renal failure	25 (17.6)	52 (17.3)	0.944
Congestive heart failure	8 (5.6)	25 (8.3)	0.313
Chronic heart disease	18 (12.7)	49 (16.3)	0.317
Chronic lung disease	11 (7.7)	12 (4)	0.098
History of Stroke	11 (7.7)	27 (9)	0.661
Charleston comorbidity score Median (IQR 25.75)	1 (0.3)	1 (0.3)	0.931
Fever > 38.2	67 (47.2)	155 (51.7)	0.379
Presented with Hypotension MAP < 70	11 (7.7)	17 (5.7)	0.402
Presented with WBC > 10,000	65 (45.8)	131 (43.7)	0.677
Bacteremia on presentation same pathogen as urine	6 (4.1)	16 (5.3)	0.649
Bacteremia on presentation different pathogen	1(0.6)	3 (1%)	0.998

**Table 3 jcm-11-07014-t003:** Microbiological data of urine cultures comparing cultures reported as low growth to those reported as positive.

	Reported Low Growth *n* = 145 *n* (%)	Reported Positive *n* = 300 *n* (%)	*p* Value
*Escherichia coli*	84 (58)	192 (64)	0.128
*Klebsiella* spp.	17 (12)	57 (19)	0.057
*Enterococcus* spp.	19 (13)	12 (4)	<0.001
*Pseudomonas aeruginosa*	13 (9)	12 (4)	0.028
*Proteus mirabilis*	8 (6)	16 (5)	0.999
*Morganella morganii*	4 (2)	11 (4)	0.999

**Table 4 jcm-11-07014-t004:** Comparison of the antibiotic treatment rates on low-growth vs. positive-growth group and hard outcome of untreated patients in both groups.

	LOW Growth (%)	Positive Growth (%)	
Antibiotic treatment	100(69%)	244(81.3%)	*p* = 0.015
Not treated	45(31%)	56(18.7%)	*p* = 0.015
**Not treated** recurrent hospitalization	1(2.2%)	7(12.5%)	*p* = 0.057
**Not treated** Bacteremia within 30 d	4(8.8%)	6(10.7%)	*p* = 0.756
**Not treated** mortality	3(6.6%)	5(8.9%)	*p* = 0.418

**Table 5 jcm-11-07014-t005:** Comparison of patient outcomes with urine cultures that were reported as low growth to patients that were reported as positive.

	Reported Low Growth *n* = 145 *n* (%)	Reported Positive *n* = 300 *n* (%)	*p* Value
Admission duration median (IQR 25.75)	3 (0.7)	3 (0.6)	0.879
No antibiotics therapy	45 (31)	56 (18.7)	0.015
Antibiotic therapy duration median (IQR 25.75)	5 (0.9)	6 (0.9)	0.015
Recurrent admission 30 day	18 (12.7)	39 (13)	0.924
Pyelonephritis within 30 day	0 (0)	3 (1)	0.764
Positive urine culture 90-day same pathogen	6 (4.2)	12 (4)	1
Positive urine culture 90-day different pathogen	4 (2.7)	10 (3.3)	0.785
Bacteremia within 30 day	1 (0.6)	2 (0.7)	0.507
Mortality all cause 90 day	8 (5.6)	22 (7.3)	0.507
*C. difficle*	0 (0)	0 (0)	

## Data Availability

The data presented in this study are available on request from the corresponding author. The data are not publicly available due to institutional restrictions.

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
