# Peer review of "The Safety and Impact of Raising the Urine Culture Reporting Threshold in Hospitalized Patients"

_jcm, 2022, doi:10.3390/jcm11237014_

Round 1

Reviewer 1 Report

Do the authors know how often or did  the clinicians who received the Low grade growth result for a culture - ask for the full bacteriological result and antibiotic sensitivity and when/why?

in the Abx treated 100 (69%)  patients in th LG group -  how was antibiotic treatment chosen?

please report separately and compare what happened to the non treated patients in both groups (45 vs 56 pts). was there a difference in outcomes there? there lies the main difference between the groups. if there is no difference there then the basic assumption about the number of bacteria in the culture effect on outcome is wrong.

Reviewer 2 Report

this is an interesting study about the interpretation of urine cultures, however, it could be improved through some points:

- the threshold definition of ASBU according to IDSA  is 10*5 and not 10*4, it should be clearly explained in the text

- the threshold of 10*4 was adopted by the laboratory without taking into account the species (E. coli, Enterococcus and Pseudomonas....) and the number of the species?

- the presence of pyuria should be mentioned in the study and included as a parameter for the comparison of the antibiotic administration and according to the threshold (the pyuria also is a clear parameter for the treatment)

- the presence of infection signs  also should be used as a parameter for the comparison in the antibiotic administration according to the two threshold

- all the cases that you deal with were monobacterial, are there any polymicrobial cases? if it was a criterion of exclusion, please add it or explain.

Round 2

Reviewer 1 Report

The results and discussion should clarify that most of the patients in both groups (81% and 69%) were actually treated and that therefore the conclusion to be drawn is that a prospective controlled randomized study  where LG will  be reported and not treated and PG will be treated, is needed to answer this question.

I suggest adding a table as added

LOW Growth (%)

POSITIVE Growth (%)

100 (69)

244 (81)

Treated

45 (31)

56 (18.7)

Not Treated

145

300

Total
